# Stereospecific synthesis of silicon-stereogenic optically active silylboranes and general synthesis of chiral silyl Anions

Xihong Wang [1], Chi Feng[2], Julong Jiang[3], Satoshi Maeda [1,3], Koji Kubota [1,2] ✉ & Hajime Ito [1,2] ✉

Silicon-stereogenic optically active silylboranes could potentially allow the formation of chiral silyl nucleophiles as well as the synthesis of various chiral silicon compounds. However, the synthesis of such silicon-stereogenic silylboranes has not been achieved so far. Here, we report the synthesis of silicon-stereogenic optically active silylboranes via a stereospecific Pt(PPh$_3$)$_4$-catalyzed Si−H borylation of chiral hydrosilanes, which are synthesized by stoichiometric and catalytic asymmetric synthesis, in high yield and very high or perfect enantiospecificity (99% *es* in one case, and >99% *es* in the others) with retention of the configuration. Furthermore, we report a practical approach to generate silicon-stereogenic silyl nucleophiles with high enantiopurity and configurational stability using MeLi activation. This protocol is suitable for the stereospecific and general synthesis of silicon-stereogenic trialkyl-, dialkylbenzyl-, dialkylaryl-, diarylalkyl-, and alkylary benzyloxy-substituted silylboranes and their corresponding silyl nucleophiles with excellent enantiospecificity (>99% *es* except one case of 99% *es*). Transition-metal-catalyzed C−Si bond-forming cross-coupling reactions and conjugate-addition reactions are also demonstrated. The mechanisms underlying the stability and reactivity of such chiral silyl anion were investigated by combining NMR spectroscopy and DFT calculations.

The asymmetric synthesis of optically active compounds with stereogenic carbon centers is a major topic in contemporary organic synthesis and has advanced significantly over the last several decades[1,2]. Compared to the synthesis of chiral carbon compounds, synthetic routes to chiral compounds with a stereogenic silicon center, which is a higher homologue of carbon, remain less explored[3–67]. After many pioneering studies, chiral organosilicon compounds that bear silicon stereocenters have recently shown attractive and widespread application prospects in organic synthesis[5–13], materials science[14–19], medicinal chemistry[20–24], and polymer chemistry[25,26] due to their unique electronic and physical properties. The development of synthetic

methods for such silicon-stereogenic optically active compounds has also become an important research subject[27–32]. In addition to classical methods that employ stoichiometric chiral auxiliaries[33–41], transition-metal catalyzed and organocatalyzed desymmetrization of prochiral silicon-containing molecules also represents an efficient approach for the synthesis of silicon-stereogenic chiral molecules[42–67].

Despite the recent rapid ascent of chiral silicon chemistry, the development of silyl nucleophiles with stereogenic silicon centers is still not well developed. Silyllithiums are generally useful synthetic intermediates toward various organosilicon compounds, albeit that originally, the scope of silicon anions that were synthetically accessible

[1]Institute for Chemical Reaction Design and Discovery (WPI-ICReDD), Hokkaido University, Sapporo, Hokkaido 001-0021, Japan. [2]Division of Applied Chemistry, Graduate School of Engineering, Hokkaido University, Sapporo, Hokkaido 060-8628, Japan. [3]Department of Chemistry, Faculty of Science, Hokkaido University Sapporo, Hokkaido 060-0815, Japan. ✉e-mail: kbt@eng.hokudai.ac.jp; hajito@eng.hokudai.ac.jp

was considerably more limited than that of carbanions, and examples of silyl anions that can be readily synthesized still remain limited[68–77]. Although enantiomerically pure silyllithiums that show configurational stability can be used as useful silicon-stereogenic silyl group transfer reagents[78–91], synthetic access to such chiral silyllithiums remains restricted, and few successful examples have been reported to date. Sommer[78], Kawakami[79], and Strohman[80–83] have made significant contributions in this direction, showing that the reaction of chiral disilanes and lithium metal generates chiral silyllithiums with unchanged configuration via the reduction of Si–Si or Si–Ph bonds (Fig. 1A-a). Similarly, lithium metal selectively cleaves the Si–Ge bond in optically active silylgermane to yield enantiomerically pure silyllithium without racemization (Fig. 1A-a)[84]. Kawakami has obtained silyllithium in a stereo-retentive manner via tin–lithium exchange of enantiomerically pure silylstannane derivative (Fig. 1A-b)[79,85–87]. However, these three methods generate stoichiometric quantities of undesired side products, including achiral silyllithium, phenyl lithium, trimethylgermyl lithium, and trimethylstannyl lithium, which compete in the nucleophilic reaction of the chiral silyllithium and decrease their synthetic utility for the generation of chiral organosilicon compounds. In 1976, Corriu developed a cobalt–lithium exchange system in which a chiral silyllithium was partially racemized (Fig. 1A-c)[88,89]. Enantiomerically pure chlorosilanes undergo significant racemization with lithium metal or di-*tert*-butylbiphenylide (LiDBB) due to chloride-induced racemization, as reported by Oestreich (Fig. 1A-d)[79,90]. In 2010, Tomooka's group reported that chiral chlorosilanes react with an excess lithium 1-(dimethylamino)naphthalenide (LDMAN) to obtain chiral silyllithium

and silacarboxylic acids in a highly stereospecific manner, being attributed to sterically bulky substituents around the silicon atom (Fig. 1A-d)[91]. It should also be noted here that all these known methods require a silicon center with at least one aryl group and steric demand to stabilize the silyl anion, which severely restricts their practical applications. In addition, excess amount of reductant (Li, LiDBB, LDMAN, etc) and nucleophiles (MeLi, etc) are required in many cases. While the silyl anion configuration is relatively stable, there are currently no reliable and general methods available for generating chiral silyl anions from chiral silicon compounds[78–91]. Therefore, the development of practical and widely applicable methods that allow the synthesis of a wide variety of silicon-stereogenic nucleophiles is highly desirable, as these have great significance for the construction of chiral organosilicon compounds.

Silylboranes have become indispensable reagents for introducing silicon and boron groups into various substrates, and silylboranes can produce silyl nucleophiles in the presence of bases[92–98]. We anticipated that a diverse range of optically active silyl anions could be generated by employing optically active silylboranes with a silicon stereogenic center. However, silicon-stereogenic optically active silylboranes have not yet been synthesized[99–117]. Our group has developed a general synthetic method for silylboranes based on the platinum- or rhodium-catalyzed borylation of hydrosilanes, which provides access to dialkylarylsilylboranes and trialkylsilylboranes with various alkyl groups (Fig. 1B)[116]. Moreover, we have demonstrated that the preparation of trialkylsilyllithium species from the corresponding trialkylsilylboranes is feasible. Subsequently, we developed an iridium- or nickel-catalyzed

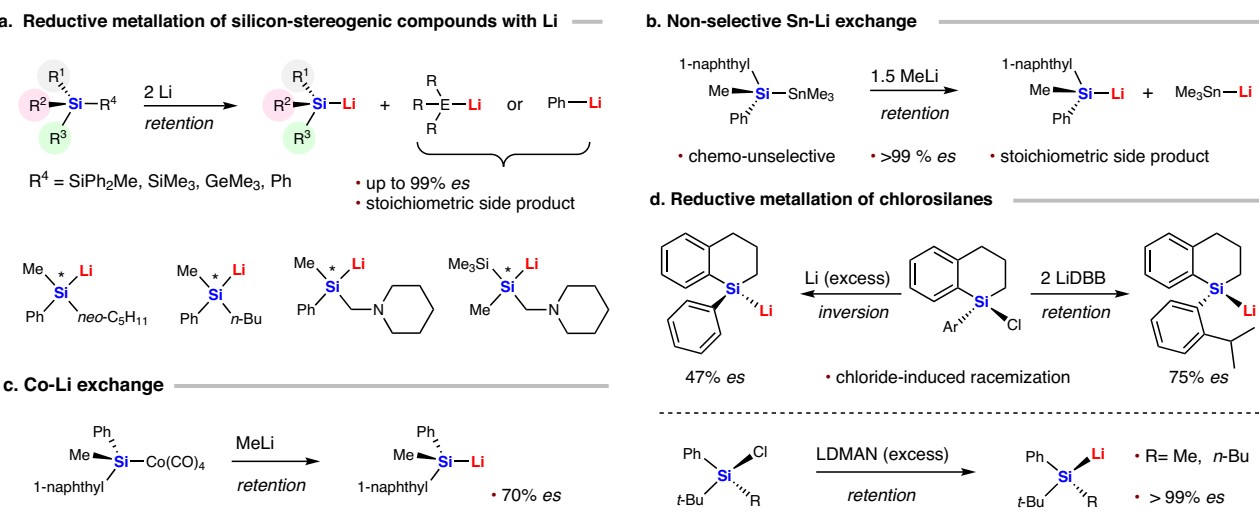

**A Reported synthetic routes to silicon-stereogenic optically active silyllithiums**

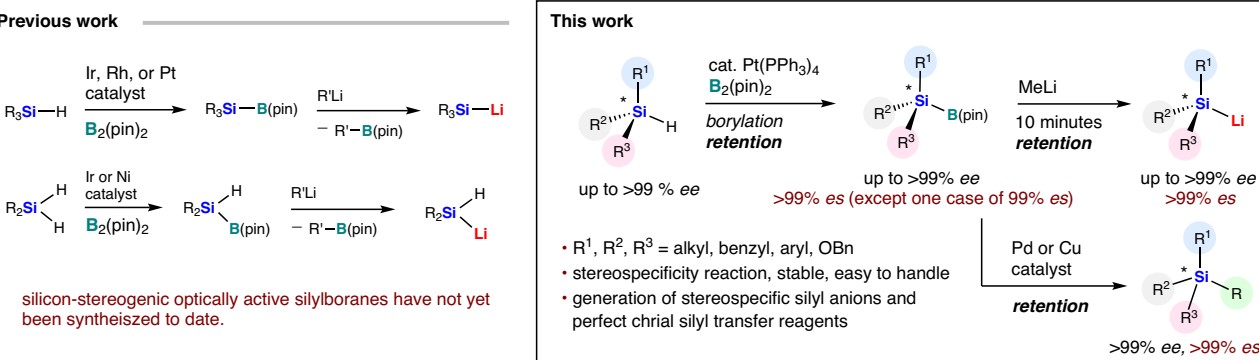

**B Synthesis of chiral silylboranes and the generation of the corresponding chiral silyl nucleophiles**

**Fig. 1 | Synthetic routes to silicon-stereogenic optically active silyllithiums.**
**A** Reported synthetic routes to silicon-stereogenic optically active silyllithiums.

**B** Synthesis of chiral silylboranes and the generation of the corresponding chiral silyl nucleophiles.

monoborylation of dihydrosilane to produce hydrosilylboronates and demonstrated the generation of dialkylhydrosilyl lithium through the activation of hydrosilylboronates with MeLi (Fig. 1B)[117] (Despite numerous attempts, asymmetric monoborylation of pro-chiral dihydrosilanes has yet to be successfully achieved). Based on our previous studies, we envisioned that the general synthesis of chiral silylboranes might be realized if the transition-metal-catalyzed Si–H borylation of the corresponding chiral hydrosilanes could proceed (Fig. 1B).

Here, we report the synthesis of silicon-stereogenic optically active silylboranes and the generation of a wide range of chiral silyl nucleophiles. We found that the Pt(PPh₃)₄-catalyzed borylation of chiral hydrosilanes, which are synthesized by stoichiometric and catalytic asymmetric synthesis, in high yield and perfect stereoselectivity with retention of their configuration (Fig. 1B). The single-crystal X-ray diffraction analyses of these chiral silylboranes are presented. The synthesized silicon-stereogenic silylboranes are easy to handle and can be used as silicon-stereogenic silyl-group transfer reagents with high enantiomeric purity, exhibiting complete enantiospecificity in the generation of chiral silyllithiums, silicon–silicon bond-forming reactions. In addition, we demonstrate that palladium(0)-catalyzed C–Si bond-forming cross-coupling reactions and copper(I)-catalyzed silyl-conjugate-addition reactions can also be conducted with perfect stereospecificity.

## Results and discussion

We started the investigation of the synthesis of silicon-stereogenic optically active silylboranes based on our previous research[116]. The most important aspect of this study is to unambiguously determine the absolute configuration of the hydrosilane, the silylborane, and the silyl anion. For that purpose, we first searched for a chiral crystalline hydrosilane that can be analyzed by single-crystal X-ray diffraction analysis. Ideally, the silylborane produced by the hydrosilane should be crystalline as well. Moreover, to determine the stereochemistry of the silyllithium, crystalline products should be obtained by trapping the silyllithium without loss of the stereochemistry. Although this was a very difficult task, we finally settled on the chiral crystalline hydrosilane  (−)-(R)-[(1,1′-biphenyl)–4-yl](cyclohexyl)methylsilane  [(−)-(R) −1a], which was prepared with >99% ee by resolution of the corresponding racemic hydrosilane using preparative HPLC equipped with a chiral column (Fig. 2A; for details, see the Supplementary Information)[118]. The absolute configuration of (−)-(R)-1a was unequivocally determined via single-crystal X-ray diffraction analysis (for details, see the Supplementary Information). After screening potential catalysts for the borylation of (−)-(R)-1a (for details, see the Supplementary Information), we found that Pt(PPh₃)₄ is suitable for the borylation of (−)-(R)-1a and bis(pinacolato)diboron, which affords (−)-(R)-[(1,1′-biphenyl)-4-yl](cyclohexyl)methyl(4,4,5,5-tetramethyl-1,3,2-dioxaborolan−2-yl)silane [(−)-(R)-2a] in 73% yield with perfect enantiospecificity (>99% ee; >99% es). A single-crystal X-ray diffraction analysis of the product unambiguously confirmed the reaction proceeded with retention of the configuration of the silicon-stereogenic center (Fig. 2A; for details, see the Supplementary Information).

We then investigated the reaction of the chiral silylborane (−)-(R)-2a with chlorotriphenylsilane in the presence of methyl lithium as a nucleophilic activator of the silylboranes in order to observe whether the synthesized chiral silylborane could act as a silicon-stereogenic silyl-group transfer reagent (Fig. 2B). When (−)-(R)-2a (>99% ee) was treated with methyl lithium in THF at −78 °C for 10 min, it subsequently reacted with chlorotriphenylsilane to furnish (+)-(S)-1-[(1,1′-biphenyl) −4-yl]-1-cyclohexyl-1-methyl-2,2,2-triphenyldisilane [(+)-(S)-3a] in 85% yield with perfect enantiospecificity (>99% ee; >99% es). A single-crystal X-ray diffraction analysis confirmed the absolute configuration to be (+)-(S)-3a; the nucleophilic reaction, therefore proceeds with retention of the configuration (Fig. 2B; for details, see the Supplementary Information).

Subsequently, we performed in situ ¹¹B{¹H} NMR and ²⁹Si{¹H} NMR experiments to explore key intermediates in the reaction of (±)-2a with methyl lithium (for details, see the Supplementary Information). Kawachi and Tamao reported the formation of Ph₃SiLi (²⁹Si: δ −9.1) via a boron–lithium exchange reaction between triphenylsilylborane and methyl lithium[119,120]. Previous work by our group showed that i-Pr₃SiLi (²⁹Si: δ 14.7) is the major product in the reaction of i-Pr₃Si–B(pin) and MeLi, while the i-Pr₃Si–B(pin)/MeLi ate complex (¹¹B: δ 8.2) is generated as the minor product[116]. Similarly, two new ¹¹B signals appeared when (±)-2a was treated with 1.5 equiv MeLi in THF-d₈ at −78 °C. However, in contrast to our previous results, the small signal was consistent with Me–B(pin) (¹¹B: δ 33.5), while the large signal was tentatively attributed to the (±)-2a/MeLi ate complex (¹¹B: δ 8.5). The ²⁹Si NMR spectrum showed only one peak (δ −15.5), which most likely corresponds to (±)-{[(1,1′-biphenyl)-4-yl](cyclohexyl)(methyl)silyl}lithium. The ²⁹Si signal of the (±)-2a/MeLi ate complex was not observed, probably due to dynamic processes in equilibrium and the adjacent quadrupolar boron atom[121]. We then carried out a ²⁹Si{¹H} NMR analysis at −95 °C, which revealed a broad peak (δ −17.1) that was tentatively assigned to (±)-{[(1,1′-biphenyl)-4-yl](cyclohexyl)(methyl)silyl}lithium, albeit that Si–Li coupling was not observed. These results indicate the presence of an equilibrium between (±)-{[(1,1′-biphenyl)-4-yl](cyclohexyl)(methyl) silyl}lithium and the (±)-2a/MeLi ate complex.

We then decided to investigate the stereospecificity and configurational stability of the silicon-stereogenic optically active silyl nucleophiles generated from the chiral silylboranes. As shown in Fig. 2C, the chiral silyl nucleophile can be expected to be formed first in the equilibrium between ate complex A and silyllithium intermediate B from (−)-(R)-2a (>99% ee) via treatment with methyl lithium in THF at −78 °C. The nucleophile was then quenched with 1.0 M aqueous HCl to give the corresponding chiral hydrosilanes (−)-(R)-1a in 83% yield with >99% ee (entry 1)[79]. The absolute configuration of (−)-(R)-1a remains unchanged, demonstrating that all processes proceed with retention of the configuration and perfect enantiospecificity (>99% es)[79,81,90]. Even when the reaction temperature was increased to −40 °C, (−)-(R)-1a was obtained with >99% ee, albeit that the yield decreased to 68% (entry 2), and 26% of (−)-(R)-2a was recovered (for details, see the Supplementary Information). Further increasing the reaction temperature to room temperature resulted in a decreased yield of 21%, while the enantiomeric purity remained high (entry 3, >99% ee). Only 17% of (−)-(R)-2a was recovered, and other unidentified side products were observed (for details, see the Supplementary Information). Next, we examined the nucleophilic activators for the silylboranes. When we used n-butyl lithium instead of methyl lithium, a yield of 54% was obtained with >99% ee (entry 4). With the more reactive and basic sec-butyl lithium, the yield was further reduced to only 20%, albeit the stereospecificity of the protonation remained unchanged (>99% ee) (entry 5). Interestingly, when lithium tert-butoxide was used as the nucleophile, the reaction did not proceed, even though lithium tert-butoxide has been reported to be a good activator for other silylboranes such as Me₂PhSi–B(pin) (entry 6)[122]. With potassium tert-butoxide, only an 8% yield was obtained, while the erosion of the enantiomeric purity (>99% ee) was not observed (entry 7). When the reaction was carried out with methylmagnesium bromide, the enantiomeric excess of (−)-(R)-1a was reduced to 95% and the yield was very low (6%; entry 8). When toluene was used as the solvent, the yield of (−)-(R)-1a was 32% with >99% ee (entry 9), whereas the use of n-hexane as the solvent resulted in 43% yield with >99% ee, suggesting that a less polar solvent affects the yield but not the stereoselectivity (entry 10). In addition, when (−)-(R)-2a was treated with methyl lithium in THF at −78 °C for a longer reaction time of 2 h, (−)-(R)-1a was obtained in 87% yield with >99% ee after quenching with 1.0 M aqueous HCl (entry 11). These results demonstrate the high stability of the chiral ate complex and the silyllithium with regard to stereochemistry.

In order to investigate the reaction mechanism for the reaction of silylborane and MeLi, we conducted DFT calculations at the

**A. Stereospecific borylation of (−)-(R)-1a**

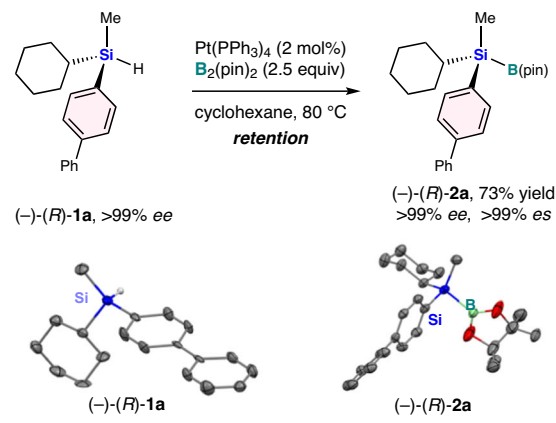

(−)-(R)-1a, >99% ee

(−)-(R)-2a, 73% yield
>99% ee, >99% es

(−)-(R)-1a

(−)-(R)-2a

**B. Stereospecific silylation of (−)-(R)-2a**

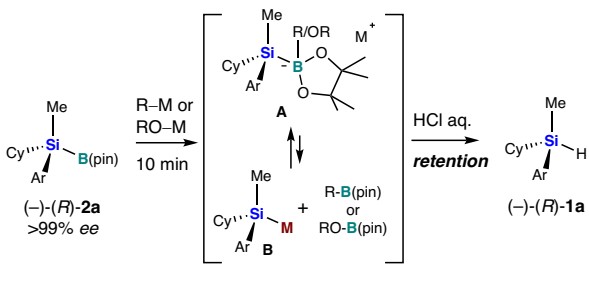

(−)-(R)-2a
>99% ee

(+)-(S)-3a, 85% yield
>99% ee, >99% es

(+)-(S)-3a

**C. Configurational stability of optically active silyl nucleophiles[a]**

(−)-(R)-2a
>99% ee

(−)-(R)-1a

| entry | activator | temp. (°C) | solvent | yield (%) | ee (%) |
|---|---|---|---|---|---|
| 1 | MeLi | −78 | THF | 83 | >99 |
| 2 | MeLi | −40 | THF | 68 | >99 |
| 3 | MeLi | rt | THF | 21 | >99 |
| 4 | n-BuLi | −78 | THF | 54 | >99 |
| 5 | s-BuLi | −78 | THF | 20 | >99 |
| 6 | t-BuOLi | −78 | THF | N.R. | - |
| 7 | t-BuOK | −78 | THF | 8 | >99 |
| 8 | MeMgBr | −78 | THF | 6 | 95 |
| 9 | MeLi | −78 | Toluene | 32 | >99 |
| 10 | MeLi | −78 | Hexane | 43 | >99 |
| 11[b] | MeLi | −78 | THF | 87 | >99 |

[a]Conditions: (−)-(R)-2a (0.1 mmol), activator (0.15 mmol), and HCl aq. (1.0 M, 200 μL) in the specified solvent (0.5 mL). The yield values refer to isolated yields. The ee values were determined by HPLC using a chiral stationary phase. [b](−)-(R)-2a with methyllithium in THF at −78 °C for 2 h.

**D. DFT calculations for the stereospecific reaction with chiral silylborane**

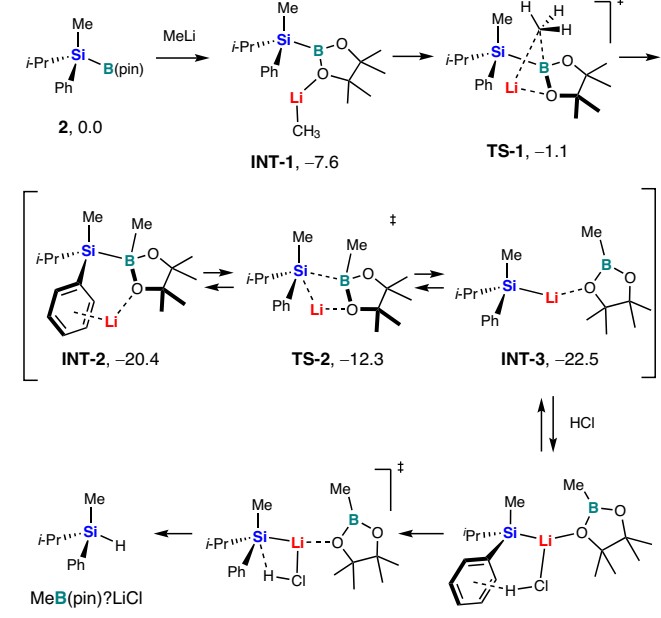

2, 0.0

INT-1, −7.6

TS-1, −1.1

INT-2, −20.4

TS-2, −12.3

INT-3, −22.5

INT-4, −27.5

TS-3, −26.4

1, −82.7

MeB(pin)?LiCl

The Gibbs free energy (kcal/mol) values are shown in parentheses.

**Fig. 2 | Synthesis of chiral silylboranes and the generation of the corresponding chiral silyl nucleophiles. A** Stereospecific borylation of chiral hydrosilane (−)-(R)-1a. **B** Stereospecific silylation of silicon-stereogenic optically active silylborane (−)-(R)-2a. **C** Configurational stability of silicon-stereogenic optically active silyl nucleophiles under various conditions. **D** DFT calculations for the stereospecific reaction between silylborane and MeLi followed by protonation. The Gibbs free energy (kcal/mol) values are shown in parentheses.

B3LYP-D3/Def2-SVP level (Fig. 2D). The reaction between model silylborane **2** and MeLi first forms complex **INT-1** (−7.6 kcal/mol), which spontaneously transforms to a thermodynamically stable boronate complex **INT-2** (−20.4 kcal/mol) through **TS-1** (−1.1 kcal/mol) with a low activation barrier (G‡ = 6.5 kcal/mol). Boronate **INT-2** proceeds via transition state **TS-2** (−12.3 kcal/mol) to form silyl lithium **INT-3** (−22.5 kcal/mol) while retaining its stereochemistry. The low activation barriers (G‡ = 8.1 and 10.2 kcal/mol) and an energy change of only −2.1 kcal/mol suggest an equilibrium between **INT-2** and **INT-3**, which is consistent with the results of the NMR experiments described above (vide supra). The reaction of **INT-3** with HCl forms **INT-4**, which is rapidly converted into protonation product **1** via **TS-3** with a very low activation barrier (G‡ = 1.1 kcal/mol) with retention of the configuration. We hypothesize that the high stereospecificity observed during the generation of silyl anions can be attributed to the avoidance of

generating silyl radicals and chloride ions, which can lead to racemization of the silicon center[90,123].

We also applied the stereospecific borylation and silylation to other silicon-stereogenic optically active hydrosilanes (Fig. 3). Most notably, (+)-(R)-methyl(naphthalen-1-yl)phenylsilane [(+)-(R)-1b] (>99% ee), which bears two aryl groups and one alkyl group[124], also reacted well in the borylation reaction to afford (+)-(R)-methyl(naphthalen-1-yl)phenyl(4,4,5,5-tetramethyl-1,3,2-dioxaborolan-2-yl)silane [(+)-(R)-2b] in 72% yield (>99% ee) (The enantiomeric purity of (+)-(R)-2b could not be evaluated on account of the poor separation on the chiral HPLC column. From (+)-(R)-1b and (−)-(S)-3b, we deduced that the enantiomeric purity of (+)-(R)-2b is >99% ee). The subsequent stereoretentive silicon−silicon bond-forming reaction of (+)-(R)-2b with benzylchlorodimethylsilane proceeded smoothly to afford (−)-(S)-1-benzyl-1,1,2-trimethyl−2-(naphthalen−1-yl)−2-phenyldisilane

**Synthesis of silicon-stereogenic optically active silylboranes and disilanes[a]**

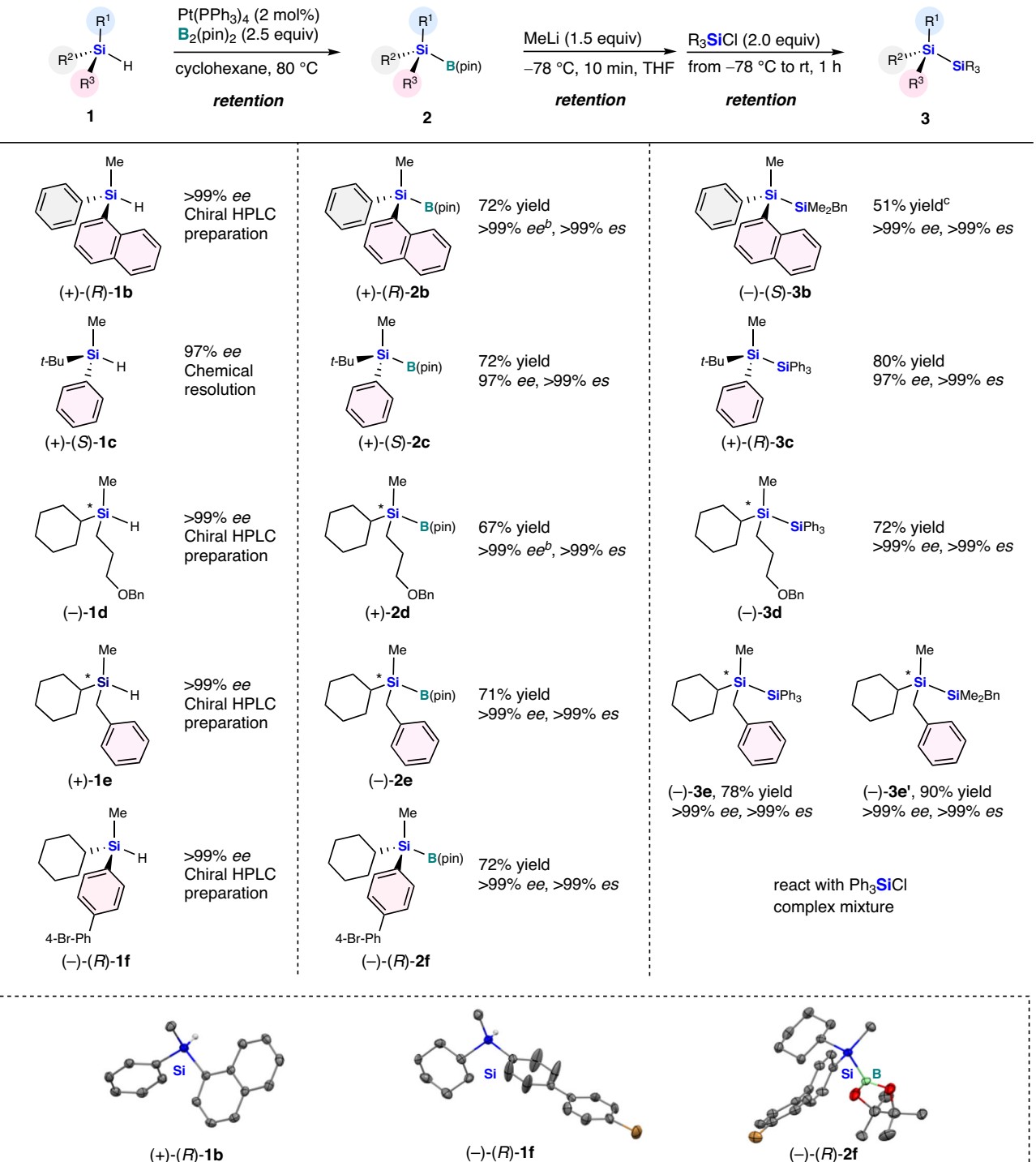

**Fig. 3 | Synthesis of silicon-stereogenic optically active silylboranes and disilanes.** [a]Conditions for the borylation system: **1** (0.3 mmol), Pt(PPh₃)₄ (2 mol%), B₂(pin)₂ (0.75 mmol) in cyclohexane (0.3 mL) at 80 °C; conditions for the Si–Si bound formation reaction: **2** (0.1 mmol), MeLi (1.2 M in Et₂O; 0.15 mmol), chlorosilane (0.2 mmol) in THF (0.5 mL). The yield values refer to isolated yields. The *ee* values were determined by chiral HPLC. [b]Presumed *ee* value. [c](+)-(*R*)-**2b** with methyllithium in THF at −78 °C for 30 min.

[(−)-(*S*)-**3b**] in moderate yield (51%) without loss of enantiomeric purity (>99% *ee*). The absolute configuration of (+)-(*R*)-**1b** was unambiguously confirmed by a single-crystal X-ray diffraction analysis after crystallization (for details, see the Supplementary Information), albeit the absolute configurations of (+)-(*R*)-**2b** and (−)-(*S*)-**3b** were deduced from the results for (+)-(*R*)-**1b**, because their single crystals suitable for XRD

analysis were not obtained. (+)-(*S*)-*tert*-Butyl(methyl)phenylsilane [(+)-(*S*)-**1c**], a chiral dialkylarylsilane, was prepared by the conventional optical resolution method reported by Oestreich[125] (We synthesized (+)-(*S*)-**1c** according to a previously reported procedure (ref. 123) and used a different diastereomer to obtain the opposite enantiomer synthesized in ref. 123). The absolute configuration of (+)-(*S*)-**1c** was

also confirmed by comparison to other reports (ref. 35)). (+)-(S)-1c was easily converted into (+)-(S)-tert-butyl(methyl)phenyl(4,4,5,5-tetramethyl-1,3,2-dioxaborolan-2-yl)silane [(+)-(S)-2c] and (+)-(R)-1-(tert-butyl)-1-methyl-1,2,2,2-tetraphenyldisilane [(+)-(R)-3c] in high yield with perfect enantioselectivity [(+)-(S)-2c: 72% yield, 97% ee; (+)-(R)-3c: 80% yield, 97% ee] (Similar to those of (+)-(R)-2b and (−)-(S)-3b, the absolute configuration of (+)-(S)-2c and (+)-(R)-3c was deduced from the above results due to the lack of their crystallinity). Trialkyl-substituted hydrosilane (−)-[3-(benzyloxy)propyl](cyclohexyl)methylsilane [(−)-1d] afforded the desired (+)-[3-(benzyloxy)propyl](cyclohexyl)methyl(4,4,5,5-tetramethyl-1,3,2-dioxaborolan-2-yl)silane [(+)-2d] in 67% yield with >99% ee. Subsequently, (+)-2d was used in the silylation reaction with chlorotriphenylsilane to give (−)-1-[3-(benzyloxy)propyl]-1-cyclohexyl-1-methyl-2,2,2-triphenyldisilane [(−)-3d] in 72% yield with >99% ee, showing perfect stereospecificity of the reaction steps. Moreover, borylation of (+)-benzyl(cyclohexyl)methylsilane [(+)-1e] provided (−)-benzyl(cyclohexyl)methyl(4,4,5,5-tetramethyl-1,3,2-dioxaborolan-2-yl)silane [(−)-2e] in 71% yield with excellent enantioselectivity (>99% ee). Then, (−)-2e was treated with chlorotriphenylsilane or benzylchlorodimethylsilane to obtain (−)-1-benzyl-1-cyclohexyl-1-methyl-2,2,2-triphenyldisilane [(−)-3e] (78% yield, >99% ee) or (−)-1,2-dibenzyl-1-cyclohexyl-1,2,2-trimethyldisilane [(−)-3e'] (90% yield, >99% ee), respectively (We did not determine the absolute configuration of (−)-1d, (+)-2d, (−)-3d, (+)-1e, (−)-2e, (−)-3e, and (−)-3e' because we could not obtain single crystals of sufficient quality for X-ray diffraction analysis). Moreover, (−)-(R)-[4'-bromo-(1,1'-biphenyl)−4-yl](cyclohexyl)methylsilane [(−)-(R)-1f] efficiently underwent the Pt(PPh₃)₄-catalyzed borylation to furnish (−)-(R)-[4'-bromo-(1,1'-biphenyl)-4-yl](cyclohexyl)methyl(4,4,5,5-tetramethyl-1,3,2-dioxaborolan-2-yl)silane [(−)-(R)-2f] in 72% yield with >99% ee. However, since the bromine substituent also reacts with methyl lithium to generate the aryl lithium species, the silicon–silicon bond-forming reaction of

(−)-(R)-2f produced a relatively complex mixture. The absolute configuration of (−)-(R)-1f and (−)-(R)-2f was unambiguously confirmed by single-crystal X-ray diffraction analysis. These results clearly demonstrate that the Pt(PPh₃)₄-catalyzed borylation of various optically active hydrosilanes proceeds stereospecifically with retention of the configuration, followed by the stereospecific generation of silyl nucleophiles and their reaction with silicon electrophiles, all of which occur in a stereoretentive manner.

The results described above clearly show that the Pt(PPh₃)₄-catalyzed borylation of the hydrosilanes proceeds stereoselectively to produce various chiral silylboranes. However, the synthesis of optically active starting hydrosilanes 1a–1f requires chiral HPLC separation or the use of a stoichiometric chiral source. To find a more efficient method, we examined the transition-metal-catalyzed preparation of chiral hydrosilanes and their stereoselective borylation (Fig. 4)[18,59]. For that purpose, we prepared chiral hydrosilane (+)-(R)-1g (94%, 97% ee; Fig. 4A) via the Rh-catalyzed enantioselective dehydrogenative Si–O coupling reported by He and conducted the borylation to obtain silylborane (+)-(R)-2g with 95% ee[18]. The low yield (34%) and slight erosion of the enantiopurity can most likely be attributed to the instability of the silyl Pt intermediate with a Si–O bond. Further reaction with MeLi and Ph₃SiCl resulted in the formation of the chiral disilane (+)-(S)-3g in high yield with complete retention of configuration (83%, 95% ee). We also synthesized optically active hydrosilane (+)-(S)-1h (90%, 96% ee) from 5 using a chiral Rh catalyst according to the procedure reported by Wang[59]. (+)-(S)-1h was subjected to a Pt-catalyzed borylation to give chiral silylborane (−)-(S)-2h in high enantiomeric excess (71%, 96% ee; Fig. 4B). These results show the high applicability of this Pt-catalyzed borylation for the highly efficient asymmetric synthesis of chiral silylboranes. To develop other efficient routes to chiral hydrosilanes, we also carried out a preliminary study on chiral ligand screening for platinum-, iridium-, nickel-, and rhodium-catalyzed asymmetric borylations of

Fig. 4 | Synthesis and reaction of silicon-stereogenic optically active silylboranes from chiral hydrosilanes prepared by catalytic asymmetric synthesis. A Rh-catalyzed enantioselective dehydrogenative Si–O coupling. B Rh-catalyzed intramolecular hydrosilylation.

**A. Pd-catalyzed silylation of 1-bromonaphthalene**

(−)-(R)-**2a**, >99% ee
1.5 equiv

Pd(PPh₃)₄ (10 mol%)
K₂CO₃ (3.0 equiv)

toluene, 120 °C, 49 h

(−)-(S)-**6a**, 42% yield
>99% ee, >99% es

(−)-(S)-**6a**

**B. Pd-catalyzed silylation of 1-(bromomethyl)naphthalene**

(−)-(R)-**2a**, >99% ee
1.2 equiv

Pd(PPh₃)₄ (4 mol%)
Ag₂O (1.0 equiv)

THF, rt, 24 h

(−)-(S)-**7a**, 76% yield
>99% ee, >99% es

(−)-(S)-**7a**

**C. Cu-catalyzed silyl conjugate addition**

(−)-(R)-**2a**, >99% ee
1.1 equiv

CuCl (10 mol%)
IMes·HCl (11 mol%)
t-BuONa (22 mol%)

MeOH (2.0 equiv)
THF, 35 °C, 26 h

(+)-**8a**, 50% yield
>99% ee, >99% es

**Fig. 5 | Transition metal-catalyzed reactions of silicon-stereogenic optically active silylboranes. A** Pd-catalyzed silylation of 1-bromonaphthalene. **B** Pd- catalyzed silylation of 1-(bromomethyl)naphthalene. **C** Cu-catalyzed silyl conjugate addition.

achiral *tert*-butyl(phenyl)silane with B₂(pin)₂, albeit that unfortunately mostly racemic products were obtained in most cases (for details, see the Supplementary Information).

Subsequently, we investigated transition-metal-catalyzed silyla-tion reactions with the silicon-stereogenic optically active silylboranes (Fig. 5A). In 2015, He's group reported a palladium-catalyzed reaction of silylboranes with aryl bromides[126]. By modifying the reaction con-ditions, the more sterically hindered (−)-(R)-**2a** was made compatible with the reaction with no erosion of the enantiomeric excess, albeit the products were obtained only in moderate yield. When we conducted the reaction of (−)-(R)-**2a** with 1-bromonaphthalene, the corresponding (−)-(S)-(1,1'-biphenyl)−4-yl(cyclohexyl)methyl(naphthalen-1-yl)silane [(−)-(S)-**6a**] was obtained in 42% yield with outstanding enantiospeci-ficity (>99% ee; >99% es). The absolute configuration of (−)-(S)-**6a** was confirmed by a single-crystal X-ray diffraction analysis, which revealed retention of the stereochemistry (Fig. 5A, for details, see the Supple-mentary Information).

We further investigated transition-metal-catalyzed reactions for the introduction of a chiral silicon group. For that purpose, we per-formed a palladium-catalyzed silylation of primary alkyl halides with silylboranes, which was reported by Xu's group in 2016[127]. The reaction between (−)-(R)-**2a** and 1-(bromomethyl)naphthalene proceeded effectively to afford the desired (−)-(S)-(1,1'-biphenyl)−4-yl(cyclohexyl)

methyl(naphthalen-1-ylmethyl)silane [(−)-(S)-**7a**] in 76% yield in a completely stereoretentive manner (>99% ee; >99% es) (Fig. 5B). The absolute configuration of (−)-(S)-**7a** was unambiguously confirmed by a single-crystal X-ray diffraction analysis (Fig. 5B, for details, see the Supplementary Information).

We next examined the utility of silicon-stereogenic optically active silylborane in copper(I)-catalyzed conjugate addition of chiral silylborane and phenyl acrylate (Fig. 5C). In 2010, Hoveyda's group reported the enantioselective conjugate silyl additions to unsaturated carbonyls catalyzed by a copper(I)/NHC system[128]. After modifying the reaction conditions, (−)-(R)-**2a** was successfully applied in this reaction, and (+)-phenyl {3-[(1,1'-biphenyl)−4-yl](cyclohexyl)(methyl)silyl}pro-panoate [(+)-**8a**] was obtained in 50% yield without loss of enantios-electivity (>99% ee; >99% es). It should furthermore be noted here that copper(I)-catalyzed conjugate reaction proceeded in a stereospecific manner, although it was not possible to determine the stereochemistry due to a lack of crystallinity of the obtained product.

Silicon-stereogenic optically active silylboranes were synthesized by a Pt(PPh₃)₄-catalyzed stereospecific borylation of chiral hydro-silanes. The corresponding chiral silyl nucleophiles generated from the chiral silylboranes are configurationally stable even at room tem-perature and react with chlorosilanes to yield the corresponding dis-ilanes in a stereoretentive manner >99% enantiospecificity). The

synthesized chiral silylboranes can also be used as silicon-stereogenic optically active silyl-group transfer reagents in various transition-metal-catalyzed silylation reactions. Combined with the recent catalytic asymmetric synthesis of chiral silanes, a wide range of chiral silylboranes will be possible in the future[50–67]. The present study can thus be expected to significantly expand the chemistry of chiral silylboranes, providing exciting opportunities to develop silicon-stereogenic optically active bioactive molecules, polymers, and optoelectronic materials.

## Methods

### Representative procedure for the platinum-catalyzed borylation of chiral hydrosilane

Chiral hydrosilane **1** (0.30 mmol, 1.0 equiv), bis(pinacolato)diboron (0.75 mmol, 2.5 equiv) were placed in a vial with a screw cap containing a Teflon®-coated rubber septum under air. The vial was placed in a glove box, and then Pt(PPh$_3$)$_4$ (0.006 mmol, 2.0 mol %) was added to the vial in the glove box under an argon atmosphere. After closing the vial, the reaction vial was removed from the glove box, and then dry cyclohexane (0.3 mL) was added to the vial via a syringe. After being stirred at 80 °C for 24 h, the reaction mixture was analyzed by GC to check the completeness of the reaction. The mixture was directly filtered through celite with Et$_2$O as an eluent, and then the resultant solution was concentrated under reduced pressure. The crude product was purified by flash column chromatography (SiO$_2$, hexane/Et$_2$O, 100:0 to 99:1) to give the corresponding product **2**.

### Representative procedure for activation of chiral silylborane with methyl lithium followed by reaction with chlorosilane

Chiral silylborane **2** (0.10 mmol, 1.0 equiv) was placed in a vial with a screw cap containing a Teflon®-coated rubber septum. After the vial was connected to manifold with nitrogen and a vacuum line through a needle, it was evacuated and backfilled with nitrogen. This cycle was repeated three times. Dry THF (0.5 mL) was added to the vial through the rubber septum using a syringe, and the mixture was cooled to −78 °C. Then, MeLi (0.15 mmol, 1.5 equiv) was added to the vial. After the mixture was stirred at −78 °C for 10 min, chlorosilane (0.20 mmol, 2.0 equiv) was added dropwise to the vial at −78 °C. The mixture was allowed to warm to room temperature and stirred for 1 h. After that, the mixture was quenched by the addition of EtOH and filtered through a short silica-gel column with Et$_2$O as an eluent, then the resultant solution was concentrated under reduced pressure. The crude product was purified by flash column chromatography (SiO$_2$, hexane/Et$_2$O, 100:0 to 99:1) and then further purified by GPC to give the corresponding product **3**.

## Data availability

The data reported in this paper are available in the main text or the Supplementary Information. Raw data are also available from the corresponding author on request. Materials and methods, experimental procedures, characterization data, $^1$H, $^{13}$C, $^{29}$Si, $^{11}$B NMR spectra, HPLC chromatograms and mass spectrometry data are available in the Supplementary Information. Crystallographic data for the structures reported in this Article have been deposited at the Cambridge Crystallographic Data Centre, under deposition numbers CCDC 2193705, 2193706, 2193707, 2193708, 2193709, 2193710, 2193711, 2193712, and 2244288. Copies of the data can be obtained free of charge via www.ccdc.cam.ac.uk/data_request/cif.

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

## Acknowledgements

This work was supported by the Japan Society for the Promotion of Science (JSPS) via KAKENHI grants 22H00318 (H.I.), 21H01926 (K.K.), 22H05328 (K.K.), and 22K18333 (H.I.) as well as by the JST via CREST grant JPMJCR19R1 (H.I.) and FOREST grant JPMJFR201I (K.K.) as well as by the Institute for Chemical Reaction Design and Discovery (ICReDD), which was established by the World Premier International Research Initiative (WPI), MEXT, Japan.

## Author contributions

K.K. and H.I. conceived and designed the study. X.W., J.J., S.M., K.K., and H.I. co-wrote the paper. X.W. performed the chemical experiments and analyzed the data. C.F. measured and analyzed the single-crystal X-ray diffraction data. J.J. and S.M. conducted the theoretical study. All authors discussed the results and the manuscript.

## Competing interests

The authors declare no competing interests.
