## [Peer Review File · Nature Communications]

Stereospecific Synthesis of Silicon-Stereogenic Optically Active Silylboranes and General Synthesis of Chiral Silyl AnionsReviewers' Comments:

Reviewer #1:

Remarks to the Author:

This manuscript describes asymmetric synthesis of chiral silylboranes having stereogenic silicon center, and their stereospecific transformations. The authors developed stereospecific borylation of optically active chiral hydrosilanes promoted by platinum catalyst. The thus obtained chiral silylboranes can be converted into chiral silyllithium by a reaction with MeLi, and aryl- and alkylsilanes by a treatment with Pd and Cu catalyst, via stereospecific conversion of the boryl group. The stereochemical course of the reactions were revealed by the determination of the absolute stereochemistry of the chiral silicon molecules. These results would contribute to the development of chemistry of chiral silicon molecules. I recommend the publication of this manuscript as a communication paper of nature communications.

comments

Page 2, line 33: The authors should refer the following paper as a substantial example of "medicinal chemistry" of chiral silicon molecules.

K. Tomooka et al., *Angew. Chem. Int. Ed.* 2016, 55, 5814.

Page 2, line 50: There is a description of "--via the reduction of --- or Si-Ph bonds (Figure 1A)." in the main text, but there is no corresponding description of transformation of Si-Ph bond in Fig. 1A.

Page 2, line 50: Fig. 1A - Fig. 1A-a

Page 2, line 52: Fig. 1A - Fig. 1A-a

Page 2, line 54: Fig. 1A - Fig. 1A-b

same below

page 10, line 196, "We hypothesize that the high stereospecificity observed during the generation of silyl anions can be attributed to the avoidance of generating silyl radicals and chloride ions": The authors should refer appropriate papers as the evidence; for example "ref. 89".

Reviewer #2:

Remarks to the Author:

The construction of Silicon stereocenters is an important accomplishment. This manuscript reported the synthesis of silicon-stereogenic ailylboranes using chiral silanes and enantiopurity transformations. The chemical yields are generally good, but not broad. Although this chemistry is undoubtedly helpful for synthetic chemists, due to the following limitations, I think this study has not met the novelty standard of Nature Communications, and it is more suitable for a more specialized journal. If the authors could realize the asymmetric Si-H borylation using racemic silanes, I would like to reconsider it.

General comments:

1. The method for the construction of Si-B fragments has been well developed using the Ir, Rh, Pt, Ir and Ni complexes. More importantly, this manuscript using the same method developed by the same group (Ref 115) to access the Si-B bonds. This Si-B bonds construction method have already been reported. The major novelty for this manuscript is using the optically active silanes to give chiral silylboranes. While, based on the proposed mechanism for the Pt- and other transition-metal catalyzed silanes borylation reactions (Ref 115, *Organometallics* 2008, 27, 6013–6019, and *J. Am. Chem. Soc.* 2008, 130, 24, 7534–7535), the chiral retention of silylboranes from chiral silanes by Pt-catalyzed borylation is reasonable.
2. The synthesis of chiral silanes using HPLC with chiral column or some other group developed methods, which is not valuable for industry or academic studies. The more detailed mechanism for the reaction of silylboranes with MeLi (Ref 115, *Chem. Sci.*, 2021, 12, 11799–11804) and the

transformations of chiral silylboranes using already known methods are also not attractive (Chem. Rev. 2013, 113, 402–441. and Chem. Soc. Rev., 2021, 50, 20102073).

Some constructive suggestions:

1. The bromine substituent silylboranes give complex mixture, what's results about other functional groups?
2. The "yield" should be added in Figure 3, Figure 4 and Figure 5.
3. Please check the structure in supporting information p168 and p171 .

Reviewer #3:

Remarks to the Author:

Dear professor Hajime Ito:

Organosilicon compounds are very important, which can be used in medicine and materials science. It is a valuable subject to study the synthetic methods of chiral organosilicon compounds. Compared with the synthesis methods involving carbon-stereogenic center, the research on synthesis involving silicon-stereogenic center is very few. This paper reports the synthesis of silicon-stereogenic optically active silylboranes via Pt(PPh₃)₄-catalyzed Si-H borylation of chiral hydrosilanes with high yields and excellent enantiospecificities (>99% es except for one case of 97.5% es), and general synthesis of chiral silyl anions through reactions of chiral silylboranes and MeLi with high yields and complete retention of the configurations, the configurations of silyllithiums can be judged from the silylation or quenching with aqueous HCl. A chiral silylborane (-)-(R)-2a (>99% ee), has been successfully applied to three transition metal-catalyzed silylation reactions, affording chiral silanes with a quaternary silicon-stereogenic center with moderate to high yields and complete enantiospecificity. The silicon-stereogenic silylboranes and silyllithiums proved to be good silylating reagents with complete enantiospecificity (>99% es). This research is significant, groundbreaking and has yielded good results. Therefore, I recommend its publication in Nature Communications after the following revisions.

(1) In the main text, please list Figure 1A (a), (b), (c), (d) in appropriate positions and cite corresponding references respectively.

(2) Please provide peak area ratio in the figures of HNMR of Supporting Informations.

(3) Lines 16,17, "in high yield and perfect enantiospecificity (>99% es) with retention of the configuration" is suggested to be changed to "in high yield and very high or perfect enantiospecificity (97.5% es in one case, and >99% es in the others) with retention of the configuration" (see: page 16, figure 4A, 2g was formed from 1g with 97.5% es). Similarly, lines 21, 22, "with excellent enantiospecificity (>99% es)" is suggested to be changed to "with excellent enantiospecificity (>99% es except one case of 97.5% es)" Accordingly, it is suggested that the description of the es data for the formation of silylboranes in Figure 1B (this work) be changed to ">99% es (except one case of 97.5% es)". (See the scheme 1 below)

(4) Lines 260, 261, 262, "The decreased yield (34%) and slight erosion of the stereoselectivity can most likely be attributed to the instability of the silyl Pt intermediate with a Si-O bond", suggest changing to "The low yield (34%) and slight erosion of the enantiopurity can most likely be attributed to the instability of the silyl Pt intermediate with a Si-O bond."

(5) Lines 262, 263, "Further reaction with MeLi and Ph₃SiCl resulted in the chiral disilane (+)-(S)-3g in high yield with perfect stereoselectivity (83%, 95% ee)", suggest changing to "Further reaction with MeLi and Ph₃SiCl resulted in the formation of the chiral disilane (+)-(S)-3g in high yield with complete retention of configuration (83%, 95% ee)."

sincerely,

Ming Zhang

This work

suggest changing to
">99% es
(except one case of 97.5% es)"

Scheme 1

Reviewer: 1

Comments:

This manuscript describes asymmetric synthesis of chiral silylboranes having stereogenic silicon center, and their stereospecific transformations. The authors developed stereospecific borylation of optically active chiral hydrosilanes promoted by platinum catalyst. The thus obtained chiral silylboranes can be converted into chiral silyllithium by a reaction with MeLi, and aryl- and alkylsilanes by a treatment with Pd and Cu catalyst, via stereospecific conversion of the boryl group. The stereochemical course of the reactions were revealed by the determination of the absolute stereochemistry of the chiral silicon molecules. These results would contribute to the development of chemistry of chiral silicon molecules. I recommend the publication of this manuscript as a communication paper of nature communications.

Comment 1-1:

Page 2, line 33: The authors should refer the following paper as a substantial example of "medicinal chemistry" of chiral silicon molecules.

K. Tomooka et al., *Angew. Chem. Int. Ed.* 2016, 55, 5814.

Response 1-1:

We regret that this important paper had not been cited in the original manuscript. It has been added to the revised manuscript as follows.

Before:

After many pioneering studies, chiral organosilicon compounds that bear silicon stereocenters have recently shown attractive and widespread application prospects in organic synthesis,⁵⁻¹³ materials science,¹⁴⁻¹⁹ medicinal chemistry,²⁰⁻²³ and polymer chemistry^{24,25} due to their unique electronic and physical properties.

After:

After many pioneering studies, chiral organosilicon compounds that bear silicon stereocenters have recently shown attractive and widespread application prospects in organic synthesis,⁵⁻¹³ materials science,¹⁴⁻¹⁹ medicinal chemistry,²⁰⁻²⁴ and polymer chemistry^{25,26} due to their unique electronic and physical properties.

24. Igawa, K., Yoshihiro, D., Abe, Y., Tomooka, K. Enantioselective synthesis of silacyclopentanes. *Angew. Chem., Int. Ed.* **55**, 5814–5818 (2016).

Comment 1-2:

Page 2, line 50: There is a description of "--via the reduction of --- or Si-Ph bonds (Figure 1A)." in the main text, but there is no corresponding description of transformation of Si-Ph bond in Fig. 1A.

Response 1-2:

Thank you for the comment. We have modified Figure 1A as shown below.

Before:

A Reported synthetic routes to silicon-stereogenic optically active silyllithiums

a. Reductive metallation of disilanes or silagermane with Li

c. Co-Li exchange

b. Non-selective Sn-Li exchange

d. Reductive metallation of chlorosilanes

After:

A Reported synthetic routes to silicon-stereogenic optically active silyllithiums

a. Reductive metallation of silicon-stereogenic compounds with Li

c. Co-Li exchange

b. Non-selective Sn-Li exchange

d. Reductive metallation of chlorosilanes

Comment 1-3:

Page 2, line 50: Fig. 1A - Fig. 1A-a

Page 2, line 52: Fig. 1A - Fig. 1A-a

Page 2, line 54: Fig. 1A - Fig. 1A-b

same below

Response 1-3:

Revised as suggested.

Comment 1-4:

page 10, line 196, "We hypothesize that the high stereospecificity observed during the generation of silyl anions can be attributed to the avoidance of

generating silyl radicals and chloride ions": The authors should refer appropriate papers as the evidence; for example "ref. 89".

Response 1-4:

Thank you very much for the suggestion. We have added appropriate references to the sentence.

Before:

We hypothesize that the high stereospecificity observed during the generation of silyl anions can be attributed to the avoidance of generating silyl radicals and chloride ions, which can lead to racemization of the silicon center.

After:

We hypothesize that the high stereospecificity observed during the generation of silyl anions can be attributed to the avoidance of generating silyl radicals and chloride ions, which can lead to racemization of the silicon center.^{90,124}

124. Chatgililoglu, C. Structural and chemical properties of silyl radicals. *Chem. Rev.* **95**, 1229–1251 (1995).

Reviewer: 2

Comments:

Comment 2-1:

The construction of silicon stereocenters is an important accomplishment. This manuscript reported the synthesis of silicon-stereogenic silylboranes using chiral silanes and enantiopurity transformations. The chemical yields are generally good, but not broad. Although this chemistry is undoubtedly helpful for synthetic chemists, due to the following limitations, I think this study has not met the novelty standard of Nature Communications, and it is more suitable for a more specialized journal. If the authors could realize the asymmetric Si–H borylation using racemic silanes, I would like to reconsider it.

Response 2-1:

The purpose of this study is to conduct fundamental research on optically active silylboron compounds, where silicon serves as the stereogenic center. Our study's key achievements include the successful synthesis of chiral silylboranes, which can be considered a groundbreaking accomplishment. Moreover, we have demonstrated that the derivatization of these chiral silylboranes, involving the generation of chiral silyl anions as well as Pd- and Cu-catalyzed reactions, proceed with almost perfect stereospecificity (retention). Furthermore, we have unequivocally determined the stereochemistry for all critical compounds and reactions using single-crystal X-ray crystallography (nine new compounds). This not only requires great efforts but represents a crucial contribution to this field.

The reviewer's comments primarily focus on the potential application of chiral organosilanes in organic synthesis, rather than on the fundamental significance of our study. Accordingly, we would like to argue that this reviewer's criticism is, although in parts arguably correct, somewhat imbalanced.

We agree that the development of asymmetric Si–H borylation reactions would provide an attractive route to chiral silylboranes and improve their practical utility in organic synthesis. Therefore, we have carried out a preliminary study on chiral ligand screening for transition-metal-catalyzed asymmetric borylations of achiral *tert*-butyl(phenyl)silane with B₂(pin)₂. We tested a variety of representative chiral ligands for platinum-, iridium-, nickel-, and rhodium-catalyzed borylation reactions, albeit that unfortunately mostly racemic products were obtained in most cases (see below). We will continue this challenging research in order to develop a high-performance asymmetric catalytic route for the Si–H borylation of prochiral dihydrosilanes as an independent follow-up project.

Screening of chiral ligands for Pt-catalyzed borylation

Screening of chiral ligands for Ir-catalyzed borylation

Screening of chiral ligands for Ni-catalyzed borylation

Screening of chiral ligands for Rh-catalyzed borylation

Comment 2-2:

1. The method for the construction of Si-B fragments has been well developed using the Ir, Rh, Pt, Ir and Ni complexes. More importantly, this manuscript using the same method developed by the same group (Ref 115) to access the Si-B bonds. This Si-B bonds construction method have already been reported. The major novelty for this manuscript is using the optically active silanes to give chiral silylboranes. While, based on the proposed mechanism for the Pt- and other transition-metal catalyzed silanes borylation reactions (Ref 115, *Organometallics* 2008, 27, 6013–6019, and *J. Am. Chem. Soc.* 2008, 130, 24, 7534–7535), the chiral retention of silylboranes from chiral silanes by Pt-catalyzed borylation is reasonable.

Response 2-2:

As mentioned earlier, this paper does not focus on the development of new (catalytic) reactions, but rather on the fundamental investigation of chiral silylboranes with a silicon stereogenic center. It has been widely recognized that the stereoselectivity of transition-metal-mediated reactions that involve hydrosilanes depends on the specific reaction and catalysis (*cf.* Doyle *et al.*, *J. Org. Chem.* **1990**, 55, 6082–6086; Sommer, L. H. *et al.*, *J. Am. Chem. Soc.* **1967**, 89, 1521–1522). We have employed a catalyst system similar to that used in our previous papers for synthesizing achiral silylboranes. However, the stereospecificity of the Pt-catalyzed borylation of hydrosilanes has not yet been investigated, and our present study demonstrates, for the first time, its almost perfect stereoretentive nature. This represents a significant achievement in this area of research.

Comment 2-3:

2. The synthesis of chiral silanes using HPLC with chiral column or some other group developed methods, which is not valuable for industry or academic studies.

Response 2-3:

HPLC separation of racemic compounds using a chiral stationary phase has been established as a standard procedure that is routinely employed in academic and industrial settings. Moreover, we highlight the significance of our method by demonstrating that chiral hydrosilanes obtained from asymmetric catalysis can serve as valuable starting materials for the production of chiral silylboranes.

Comment 2-4:

The more detailed mechanism for the reaction of silylboranes with MeLi (Ref 115, *Chem. Sci.*, 2021, 12, 11799–11804) and the transformations of chiral silylboranes using already known methods are also not attractive (*Chem. Rev.* 2013, 113, 402–441. and *Chem. Soc. Rev.*, 2021, 50, 20102073).

Response 2-4:

The remarkable finding in our paper is the first generation of chiral silyl anions through the reaction with MeLi. Importantly, this process preserves the stereochemistry of the silyl anion during its subsequent reaction with electrophiles. These findings thus represent significant progress in the field of fundamental organosilicon chemistry. To demonstrate the reactivity and selectivity of the silylboranes, we also employ established transformation reactions.

Comment 2-5:

1. The bromine substituent silylboranes give complex mixture, what's results about other functional groups?

Response 2-5:

We did not test other functional groups for the chiral silylboranes, albeit that we expect the functional-group compatibility to be similar to that of our previous established silylborane/base chemistry (*cf. J. Am. Chem. Soc.* **2020**, *142*, 14125; *J. Am. Chem. Soc.* **2012**, *134*, 19997).

Comment 2-6:

2. The "yield" should be added in Figure 3, Figure 4 and Figure 5.

Response 2-6:

Thank you very much for the comment. We have added the "yield" to Figures 3-5.

Before (Figure 3):

Synthesis of silicon-stereogenic optically active silylboranes and disilanes^a

After (Figure 3):

Synthesis of silicon-stereogenic optically active silylboranes and disilanes^a

Before (Figure 4):

A. Rh-catalyzed enantioselective dehydrogenative Si–O coupling

B. Rh-catalyzed intramolecular hydrosilylation

After (Figure 4):

A. Rh-catalyzed enantioselective dehydrogenative Si–O coupling

B. Rh-catalyzed intramolecular hydrosilylation

Before (Figure 5):

A. Pd-catalyzed silylation of 1-bromonaphthalene

B. Pd-catalyzed silylation of 1-(bromomethyl)naphthalene

C. Cu-catalyzed silyl conjugate addition

After (Figure 5):

A. Pd-catalyzed silylation of 1-bromonaphthalene

B. Pd-catalyzed silylation of 1-(bromomethyl)naphthalene

C. Cu-catalyzed silyl conjugate addition

Comment 2-7:

3. Please check the structure in supporting information p168 and p171 .

Response 2-7:

Thank you very much for the comment. We have modified the structures shown in the HPLC data as below.

Before:

Chromaster System Manager Report

Analyzed Date and Time: 2023/02/16 22:03 Reported Date and Time: 2023/02/17 09:41:23

Processed Date and Time: 2023/02/17 09:41

Data Path: C:\WIN32APP\CHROMASTER\WXH\DATA\0501\

Processing Method: IB_UV

System (acquisition): Sys 1

Series: 0501

Application(data): WXH

Vial Number: 11

Sample Name: wxh-411-IB-0%IPA

Vial Type: UNK

Injection from this vial: 1 of 1

Volume: 10.0 ul

Sample Description:

Chrom Type: Fixed WL Chromatogram, 268 nm

Processing Method: IB_UV

Method Developer:

Pump 1: 5110

Pump 1 Solvent A: hexane

Pump 1 Solvent B: 2-propanol

Pump 1 Solvent C:

Pump 1 Solvent D:

Method Description:

Chrom Type: Fixed WL Chromatogram, 268 nm

Peak Quantitation: AREA

Calculation Method: AREA%

No.	RT	Area	Conc 1	BC
1	8.700	2476646	47.601	MC
2	9.387	2726303	52.399	MC
		5202949	100.000	

Peak rejection level: 0

After:

CSM: WXH Series: 0501 Report Name: modified System: Sys 1

Chromaster System Manager Report

Analyzed Date and Time: 2023/02/16 22:03 Reported Date and Time: 2023/06/26 14:55:34

Processed Date and Time: 2023/06/26 14:55

Data Path: C:\WIN32APP\CHROMASTER\WXH\DATA\0501\

Processing Method: column1(IA-3)_0.3%

System (acquisition): Sys 1

Series: 0501

Application (data): WXH

Vial Number: 11

Sample Name: wxh-411-IB-0%IPA

Vial Type: UNK

Injection from this vial: 1 of 1

Volume: 10.0 ul

Sample Description:

Chrom Type: Fixed WL Chromatogram, 268 nm

Processing Method: column1(IA-3)_0.3%

Method Developer:

Pump 1: 5110

Pump 1 Solvent A: hexane

Pump 1 Solvent B: 2-propanol

Pump 1 Solvent C:

Pump 1 Solvent D:

Method Description:

Chrom Type: Fixed WL Chromatogram, 268 nm

Peak Quantitation: AREA

Calculation Method: AREA%

No.	RT	Area	Conc 1	BC
1	8.700	2476646	47.601	MC
2	9.387	2726303	52.399	MC
		5202949	100.000	

Peak rejection level: 0

Before:

D-2000: Isocratic Series: 3882 Report Name: modified System: Sys 1
c HPLC

D-2000 Elite HPLC System Manager Report

Analyzed Date and Time: 2022/02/22 11:12 Reported Date and Time: 2022/02/22 16:51

Processed Date and Time: 2022/02/22 16:51

Data Path: C:\WIN32APP\D2000HSM\Isocratic\DATA\3882\

Processing Method: 0.0/100.0 iPrOH/Hexane

System (acquisition): Sys 1 Series: 3882

Application(data): Isocratic HPLC Vial Number: 182

Sample Name: WXH-228-OD-0% Vial Type: UNK

Injection from this vial: 1 of 1 Volume: 10.0 ul

Sample Description:

Chrom Type: HPLC Channel : 1

Processing Method: 0.0/100.0 iPrOH/Hexane

Column Type: OD-H 2

Method Developer: Administrator

Pump A: L-2130

Pump A Solvent A: Hexane

Pump A Solvent B: 10/90 iPrOH/Hexane

Pump A Solvent C: iPrOH

Pump A Solvent D: EtOH

Method Description:

Chrom Type: HPLC Channel : 1

Peak Quantitation: AREA

Calculation Method: AREA%

No.	RT	Area	Area %
1	31.15	5669358	100.000
		5669358	100.000

Peak rejection level: 0

After:

D-2000: IsocratiSeries: 3882 Report Name: modified System: Sys 1
c HPLC

D-2000 Elite HPLC System Manager Report

Analyzed Date and Time: 2022/02/22 11:12 Reported Date and Time: 2022/02/22 16:51
Processed Date and Time: 2022/02/22 16:51

Data Path: C:\WIN32APP\D2000HSM\Isocratic\DATA\3882\
Processing Method: 0.0/100.0 iPrOH/Hexane
System (acquisition): Sys 1 Series: 3882
Application(data): Isocratic HPLC Vial Number: 182
Sample Name: WXH-228-OD-0% Vial Type: UNK
Injection from this vial: 1 of 1 Volume: 10.0 ul
Sample Description:

Chrom Type: HPLC Channel : 1

Processing Method: 0.0/100.0 iPrOH/Hexane
Column Type: OD-H 2 Method Developer: Administrator
Pump A: L-2130
Pump A Solvent A: Hexane Pump A Solvent B: 10/90 iPrOH/Hexane
Pump A Solvent C: iPrOH Pump A Solvent D: EtOH
Method Description:

Chrom Type: HPLC Channel : 1

Peak Quantitation: AREA
Calculation Method: AREA%

No.	RT	Area	Area %
1	31.15	5669358	100.000
		5669358	100.000

Peak rejection level: 0

Before:

D-2000: Isocratic Series: 3884 Report Name: modified System: Sys 1
c HPLC

D-2000 Elite HPLC System Manager Report

Analyzed Date and Time: 2022/02/22 15:47 Reported Date and Time: 2022/02/22 16:48

Processed Date and Time: 2022/02/22 16:48

Data Path: C:\WIN32APP\D2000HSM\Isocratic\DATA\3884\

Processing Method: 0.0/100.0 iPrOH/Hexane

System (acquisition): Sys 1 Series: 3884

Application(data): Isocratic HPLC Vial Number: 181

Sample Name: WXH-167-race-OD-0% Vial Type: UNK

Injection from this vial: 1 of 1 Volume: 10.0 ul

Sample Description:

Chrom Type: HPLC Channel : 1

Processing Method: 0.0/100.0 iPrOH/Hexane

Column Type: OD-H 2

Method Developer: Administrator

Pump A: L-2130

Pump A Solvent A: Hexane

Pump A Solvent B: 10/90 iPrOH/Hexane

Pump A Solvent C: iPrOH

Pump A Solvent D: EtOH

Method Description:

Chrom Type: HPLC Channel : 1

Peak Quantitation: AREA

Calculation Method: AREA%

No.	RT	Area	Area %	
1	25.53	11603304	50.939	
2	30.67	11175631	49.061	
			22778935	100.000

Peak rejection level: 0

After:

D-2000: Isocratic Series: 3884
c HPLC

Report Name: modified System: Sys 1

D-2000 Elite HPLC System Manager Report

Analyzed Date and Time: 2022/02/22 15:47
Reported Date and Time: 2022/02/22 16:48
Processed Date and Time: 2022/02/22 16:48

Data Path: C:\WIN32APP\D2000HSM\Isocratic\DATA\3884\
Processing Method: 0.0/100.0 iPrOH/Hexane
System (acquisition): Sys 1
Application(data): Isocratic HPLC
Sample Name: WXH-167-race-OD-0%
Injection from this vial: 1 of 1
Sample Description:

Series: 3884
Vial Number: 181
Vial Type: UNK
Volume: 10.0 ul

Chrom Type: HPLC Channel : 1

Processing Method: 0.0/100.0 iPrOH/Hexane
Column Type: OD-H 2
Pump A: L-2130
Pump A Solvent A: Hexane
Pump A Solvent C: iPrOH
Method Developer: Administrator
Pump A Solvent B: 10/90 iPrOH/Hexane
Pump A Solvent D: EtOH
Method Description:

Chrom Type: HPLC Channel : 1

Peak Quantitation: AREA
Calculation Method: AREA%

No.	RT	Area	Area %
1	25.53	11603304	50.939
2	30.67	11175631	49.061
		22778935	100.000

Peak rejection level: 0

Reviewer: 3

Comments:

Organosilicon compounds are very important, which can be used in medicine and materials science. It is a valuable subject to study the synthetic methods of chiral organosilicon compounds. Compared with the synthesis methods involving carbon-stereogenic center, the research on synthesis involving silicon-stereogenic center is very few. This paper reports the synthesis of silicon-stereogenic optically active silylboranes via Pt(PPh₃)₄-catalyzed Si-H borylation of chiral hydrosilanes with high yields and excellent enantiospecificities (>99% es except for one case of 97.5% es), and general synthesis of chiral silyl anions through reactions of chiral silylboranes and MeLi with high yields and complete retention of the configurations, the configurations of silyllithiums can be judged from the silylation or quenching with aqueous HCl. A chiral silylborane (-)-(R)-2a (>99% ee), has been successfully applied to three transition metal-catalyzed silylation reactions, affording chiral silanes with a quaternary silicon-stereogenic center with moderate to high yields and complete enantiospecificity. The silicon-stereogenic silylboranes and silyllithiums proved to be good silylating reagents with complete enantiospecificity (>99% es). This research is significant, groundbreaking and has yielded good results. Therefore, I recommend its publication in Nature Communications after the following revisions.

Comment 3-1:

(1) In the main text, please list Figure 1A (a), (b), (c), (d) in appropriate positions and cite corresponding references respectively.

Response 3-1:

Thank you very much for the suggestion. We have modified the main text accordingly. The corresponding references had already been cited in the main text.

Comment 3-2:

(2) Please provide peak area ratio in the figures of HNMR of Supporting Informations.

Response 3-2:

We have added the peak area ratio in the ¹H NMR spectra of the Supporting Information.

Comment 3-3

(3) Lines 16,17, "in high yield and perfect enantiospecificity (>99% es) with retention of the configuration" is suggested to be changed to "in high yield and very high or perfect enantiospecificity (97.5% es in one case, and >99% es in

the others) with retention of the configuration” (see: page 16, figure 4A, 2g was formed from 1g with 97.5% es). Similarly, lines 21, 22, “with excellent enantiospecificity (>99% es)” is suggested to be changed to “with excellent enantiospecificity (>99% es except one case of 97.5% es)” Accordingly, it is suggested that the description of the es data for the formation of silylboranes in Figure 1B (this work) be changed to “>99% es (except one case of 97.5% es)”. (See the scheme 1 below).

Response 3-3:

Thank you very much for the suggestion. We have calculated the es value according to the following formula:

$$\%es = \left(1 - \frac{Er_{\text{substrate}} - Er_{\text{product}}}{Er_{\text{substrate}}} \right) \times 100$$

where $Er_{\text{substrate}}$ is the ratio of the major enantiomer of the starting materials, and Er_{product} is the ratio of major enantiomer of the products.

We have made changes according to the suggestion as shown below, albeit that based on this formula, we have added 99% es instead of 97.5% es. The details of how es value were calculated have been added to the Supporting Information.

Before:

~in high yield and perfect enantiospecificity (>99% es) with retention of the configuration.

After:

~in high yield and very high or perfect enantiospecificity (99% es in one case, and >99% es in the others) with retention of the configuration.

Before:

~with excellent enantiospecificity (>99% es)

After:

~with excellent enantiospecificity (>99% es except one case of 99% es)

Before (Figure 1B):

B Synthesis of chiral silylboranes and the generation of the corresponding chiral silyl nucleophiles

After (Figure 1B):

B Synthesis of chiral silylboranes and the generation of the corresponding chiral silyl nucleophiles

Comment 3-4

(4) Lines 260, 261, 262, “The decreased yield (34%) and slight erosion of the stereoselectivity can most likely be attributed to the instability of the silyl Pt intermediate with a Si–O bond”, suggest changing to “The low yield (34%) and slight erosion of the enantiopurity can most likely be attributed to the instability of the silyl Pt intermediate with a Si–O bond.”

Response 3-4:

Thank you very much for the suggestion. We have revised the main text accordingly.

Before:

The decreased yield (34%) and slight erosion of the stereoselectivity can most likely be attributed to the instability of the silyl Pt intermediate with a Si–O bond.

After:

The low yield (34%) and slight erosion of the enantiopurity can most likely be attributed to the instability of the silyl Pt intermediate with a Si–O bond.

Comment 3-5

(5) Lines 262, 263, “Further reaction with MeLi and Ph₃SiCl resulted in the chiral disilane (+)-(S)-3g in high yield with perfect stereoselectivity (83%, 95% ee)”, suggest changing to “Further reaction with MeLi and Ph₃SiCl resulted in the formation of the chiral disilane (+)-(S)-3g in high yield with complete retention

of configuration (83%, 95% ee).”

Response 3-5:

Thank you very much for the suggestion. We have revised the main text accordingly.

Before:

Further reaction with MeLi and Ph₃SiCl resulted in the chiral disilane (+)-(S)-3g in high yield with perfect stereoselectivity (83%, 95% ee).

After:

Further reaction with MeLi and Ph₃SiCl resulted in the formation of the chiral disilane (+)-(S)-3g in high yield with complete retention of configuration (83%, 95% ee).

Other changes:

The reference numbering scheme has been adapted according to the changes during the revision.

Reviewers' Comments:

Reviewer #1:

Remarks to the Author:

This manuscript describes asymmetric synthesis of chiral silylboranes having stereogenic silicon center, and their stereospecific transformations. The reported results would contribute to the development of chemistry of chiral silicon molecules and the manuscript is properly revised. So, I recommend the publication of this manuscript as a communication paper of nature communications.

additional comment:

Reported electrophile reacted with silyllithium was limited only proton and chlorosilane. I am wondering if other electrophiles such as alkylhalide, carbonyl compounds including CO₂ were examined. If the author has such kind of experimental result even though negative information, please add it to the manuscript.

Reviewer #2:

Remarks to the Author:

I appreciate the investigation of asymmetric Si-H borylation reaction, and suggest authors to add these preliminary results in SI and one sentence in manuscript.

Reviewer #3:

Remarks to the Author:

Dear Professor Hajime Ito,

My concerns have been well addressed in the revisions, I think this article is suitable for publication in Nature Communications now.

sincerely,

Ming Zhang

Reviewer #1:

This manuscript describes asymmetric synthesis of chiral silylboranes having stereogenic silicon center, and their stereospecific transformations. The reported results would contribute to the development of chemistry of chiral silicon molecules and the manuscript is properly revised. So, I recommend the publication of this manuscript as a communication paper of nature communications.

Comment 1-1:

Reported electrophile reacted with silyllithium was limited only proton and chlorosilane. I am wondering if other electrophiles such as alkylhalide, carbonyl compounds including CO₂ were examined. If the author has such kind of experimental result even though negative information, please add it to the manuscript.

Response 1-1:

Thank you very much for the comment. Unfortunately we haven't examined the suggested carbon electrophiles. We will explore the synthetic utility of the newly synthesized chiral silyllithium as an independent follow-up project.

Reviewer #2:

Comment 2-1:

I appreciate the investigation of asymmetric Si-H borylation reaction, and suggest authors to add these preliminary results in SI and one sentence in manuscript.

Response 2-1:

Thank you very much for the suggestions. We added the preliminary results on asymmetric Si-H borylation in the Supplementary Information and the following sentence in the revised manuscript.

In Supporting Information:

Figure S13. Chiral ligand screening for platinum-catalyzed asymmetric Si-H borylation.

Figure S14. Chiral ligand screening for iridium-catalyzed asymmetric Si–H borylation.

Figure S15. Chiral ligand screening for nickel-catalyzed asymmetric Si–H borylation.

Figure S16. Chiral ligand screening for rhodium-catalyzed asymmetric Si–H borylation.

In main text:

Before:

Further reaction with MeLi and Ph₃SiCl resulted in the formation of the chiral disilane (+)-(S)-**3g** in high yield with complete retention of configuration (83%, 95% ee). We also synthesized optically active hydrosilane (+)-(S)-**1h** (90%, 96% ee) from **5** using a chiral Rh catalyst according to the procedure reported by Wang.⁵⁹ (+)-(S)-**1h** was subjected to a Pt-catalyzed borylation to give chiral silylborane (-)-(S)-**2h** in high enantiomeric excess (71%, 96% ee; Figure 4B). These results show the high applicability of this Pt-catalyzed borylation for the highly efficient asymmetric synthesis of chiral silylboranes.

After:

Further reaction with MeLi and Ph₃SiCl resulted in the formation of the chiral disilane (+)-(S)-**3g** in high yield with complete retention of configuration (83%, 95% ee). We also synthesized optically active hydrosilane (+)-(S)-**1h** (90%, 96% ee) from **5** using a chiral Rh catalyst according to the procedure reported by Wang.⁵⁹ (+)-(S)-**1h** was subjected to a Pt-catalyzed borylation to give chiral silylborane (-)-(S)-**2h** in high enantiomeric excess (71%, 96% ee; Figure 4B). These results show the high applicability of this Pt-catalyzed borylation for the highly efficient asymmetric synthesis of chiral silylboranes. To develop other efficient routes to chiral hydrosilanes, we also carried out a preliminary study on chiral ligand screening for platinum-, iridium-, nickel-, and rhodium-catalyzed asymmetric borylations of achiral tert-butyl(phenyl)silane with B₂(pin)₂, albeit that unfortunately mostly racemic products were obtained in most cases (for details, see the Supplementary Information).

Reviewer #3:

Comment 3-1:

Dear Professor Hajime Ito,

My concerns have been well addressed in the revisions, I think this article is suitable for publication in Nature Communications now.

Response 3-1:

Thank you very much for the positive comment.